

# Emodin, a rising star in the treatment of glycolipid metabolism disorders: a preclinical systematic review and meta-analysis

Yang Xiao[1], Zhixuan Zhao[1], Binqin Chen[2], Jian Sun[3], Li Wang[2], Yu Wang[1], Zheng Nan[4] and Qi Zhang[2]

[1] Faculty of Chinese Medicine, Changchun University of Chinese Medicine, Changchun, Jilin, China
[2] Department of Endocrinology and Metabolism, Shenzhen Hospital (Futian) of Guangzhou University of Chinese Medicine, Shenzhen, Guangdong, China
[3] Faculty of Basic Medicine, Changchun University of Chinese Medicine, Changchun, Jilin, China
[4] The Affiliated Hospital of Changchun University of Chinese Medicine, Changchun, Jilin, China

Corresponding authors
Zheng Nan,
nanzheng001@aliyun.com
Qi Zhang, 596058757@qq.com

## ABSTRACT

**Background**. Rhubarb has a remarkable effect on lowering blood lipid and glucose levels, and its main component, emodin, is an anthraquinone derivative. To elucidate the role and mechanism of emodin in the treatment of type 2 diabetes mellitus (T2DM) and to provide robust evidence for its clinical application, we conducted a systematic review and meta-analysis to assess the influence of emodin on T2DM animal models and the overall therapeutic effect, and further to evaluate its benefits and risks in the management of T2DM.

**Methods**. Eight databases were searched from inception to May 2023. Two reviewers extracted the data independently. SYRCLE's risk of bias tool for animal studies was used to assess the quality of articles. RevMan V.5.4 software and STATA 15.1 software were applied for data analyses. Body weight, serum insulin level (INS), fasting blood glucose (FBG), 2-hour postprandial blood glucose (2hPG, IPGTT/OGTT), insulin tolerance test (IPITT) indicators, total cholesterol (TC), triglycerides (TG), high-density lipoprotein (HDL-c), and low-density lipoprotein cholesterol (LDL-c) were used as outcome measures. Data for outcome messures presented in graphical form were extracted using GetData graphic digitizer software (version 2.26). For outcome indicators with a small number of included studies, we will conduct descriptive analyses.

**Results**. Twelve existing studies were included in this meta-analysis, and all of the studies included in this review had a low to moderate risk of bias. The results showed that emodin significantly reduced the glucose and lipid metabolism indicators and effectively lowered body weight and serum insulin levels (FBG, 2hPG(IPGTT/OGTT), IPITT, TG, TC, LDL-c, HDL-c) ($P < 0.05$).

**Conclusion**. Emodin demonstrates significant potential in treating T2DM by reducing FBG, 2hPG (IPGTT/OGTT), IPITT, TC, TG, INS, and body weight in animal models. The therapeutic mechanisms of emodin include enhancing glucose utilization in peripheral tissues, inhibiting glucosidase absorption, alleviating insulin resistance, and strengthening L-type calcium channels. Additionally, emodin shares characteristics with first-line antidiabetic drugs such as metformin, acarbose, and repaglinide, promoting insulin secretion and enhancing cellular sensitivity to insulin. Furthermore,

emodin exhibits actions similar to glucagon-likepeptide-1(GLP-1) receptor agonists, suggesting its potential for protecting target organs. Therefore, emodin is a highly promising drug with substantial research and clinical value. However, caution should be exercised due to significant heterogeneity among the studies, and results may evolve with additional research.

# INTRODUCTION

Diabetes is a metabolic disorder diseases caused by insufficient insulin secretion or impaired insulin action (*Weir, Gaglia & Bonner-Weir, 2020*). Type 2 diabetes accounts for approximately 90% of all diabetes cases, making it the most prevalent type (*Petrov & Taylor, 2022*), its incidence is rapidly increasing worldwide (*Ling, Bacos & Ronn, 2022*).The number of diabetes-related deaths rose by 3% between 2000 and 2019 in age-standardized terms (*Collaborators, 2022*). It is predicted that by 2045, there will be 783.2 million people worldwide suffering from diabetes mellitus (DM) (*Sun et al., 2022*). Additionally, it can negatively affect an individual's economic and health well-being(*Conlin et al., 2022*). Based on these data, early detection and intensive management are essential (*Bril & Cusi, 2017*). Although there are currently many drugs available for the treatment of type 2 diabetes, many patients have not fully benefited from the existing drugs due to adverse reactions, drug intolerance, *etc.* Therefore, it is urgent to explore more drugs for the treatment of T2DM (*Ma et al., 2022*; *Wang et al., 2022*).

Due to the extensive pharmacological effects of natural medicines, an increasing number of scholars have begun to pay attention to the potential of natural medicines in treating metabolic diseases (*Li et al., 2022*). Emodin, an anthraquinone derivative originating from herbs, which has multiple beneficial effects, including antibacterial, anti-inflammatory, anti-fibrotic and anti-cancer properties, these characteristics make it a highly promising natural drug (*Lv et al., 2022*; *Song et al., 2022*; *Trybus, Trybus & Król, 2022*). Studies have shown that emodin has potential therapeutic effects on T2DM (*Liu et al., 2021*; *Martorell et al., 2021*). For instance, emodin has been proven to effectively improve disorders of glucose and lipid metabolism, reduce weight, and alleviate insulin resistance (*Li et al., 2016*; *Frkic, Richter & Bruning, 2021*; *Taghvaei & Saremi, 2022*; *Feng et al., 2024*). The main mechanisms by which emodin exerts its biological effects include inhibiting the activation of various protein kinases, reducing oxidative stress, and suppressing inflammatory responses (*Cui et al., 2020*). Moreover, no obvious adverse reactions in the gastrointestinal tract were observed in animal experiments. Therefore, emodin may become a new alternative or complementary drug for the treatment of T2DM, especially suitable for patients who are intolerant to metformin and acarbose or have severe adverse reactions, as well as those with low insulin sensitivity. This will be a positive message for clinical treatment. Although emodin has been widely used in animal experiments, there is currently insufficient evidence to directly extrapolate the results of animal experiments to clinical settings. Therefore, this

study conducted a preclinical systematic review and meta-analysis to evaluate the efficacy and mechanism of action of emodin, providing a foundation for subsequent clinical trials and promoting the research and development and market launch of emodin to provide more treatment options for diabetic patients.

## MATERIALS AND METHODS

We chose to follow the PRISMA guidelines to direct the entire systematic review and meta-analysis process, the PRISMA 2020 checklist is provided in File S1, and made a detailed registration on the PROSPERO website, with the registration number CRD42022287705.

## DATABASE AND SEARCH STRATEGY

### Database

We searched eight databases: Pubmed, Web of Science, Embase, Cochrane Library, China National Knowledge Infrastructure (CNKI), WanFang database, VIP database (VIP), and China Biology Medicine (CBM).

### Search strategy

We searched using "Type 2 diabetes", "diabetes mellitus, Type 2" and "emodin", "frangula emodin", "Frangulic Acid", "Emodin, Frangula", "Rheum Emodin", "Archin" and "Animal model", "mice" and "rat" as keywords. Free text words were combined with Medical subject headings (MeSH) to retrieve all eligible studies (File S2). Two authors (Qi Zhang and Binqin Chen) independently collected all reports, during the retrieval process, the discrepancies that emerged were resolved through discussions with anther author(Yang Xiao), and the included literature was ultimately determined.. In addition, any potentially overlooked literature can be manually searched and added. Time span: January 2000 to May 2023. The languages are Chinese and English.

## INCLUSION AND EXCLUSION CRITERIA

### Inclusion criteria

(1) Published original research, including experimental animal studies (such as randomized controlled trials, non-randomized controlled trials).
(2) Diabetic animal models (rats or mice).
(3) The treatment group was given emodin (unlimited dose), while the control group received vehicle, saline or no treatment.
(4) Outcomes: FBG, 2hPG (IPGTT/OGTT), 2hPG in the IPITT, TG, TC, HDL-c, LDL-c, INS and body weight.

### Exclusion criteria

(1) Cell lines studies or clinical trials.
(2) Not diabetic model, abstracts, review articles, commentaries, and editorials.
(3) Duplicated studies.
(4) Research without the use of emodin.
(5) Exclude studies without a control group.

(6) Exclude the literature where data cannot be obtained.

## DATA EXTRACTION

The two primary authors (Zhixuan Zhao and Jian Sun) independently and carefully extracted data from all included literature. Any disagreements between them were resolved through discussion with the third author (Li Wang), ultimately reaching a consensus. The extracted data encompass four aspects:

   (1) Author name and publication year.
   (2) Basic animal information, such as species, gender, and weight.
   (3) Methods used to establish diabetes models, including streptozotocin (STZ) and the selection criteria for animals, as well as the dose of anesthetics.
   (4) Emodin treatment methods, including administration route, dose, frequency, duration, and outcome indicators.

   It's worth noting that if experimental animals received different doses of treatment, the highest dose will be chosen. In cases where results were observed at multiple time points, variables were extracted from the last time point. Data for outcome indicators presented in graphical form were extracted using GetData graphic digitizer software (version 2.26). The two authors independently extracted the data and cross-checked with each other to ensure its accuracy. The specific extraction method is detailed in the File S3.

## RISK OF BIAS ASSESSMENT

Two authors (Yu Wang and Qi Zhang) use the SYRCLE bias risk tool to assess the quality of literature. Including the following aspects: (A) sequence generation; (B) baseline characteristics; (C) allocation concealment; (D) random housing; (E) blinding (performance bias); (F) random outcome assessment; (G) blinding (detection bias); (H) incomplete outcome data; (I) selective outcome reporting; (J) other sources of bias. The results of the assessment represent low bias risk, high bias risk, and uncertainty bias risk, respectively, with "yes" "no" and "uncertainty. Then, following the discussion between the two authors, the selected studies were obtained, and any discrepancies were resolved.

## DATA SYNTHESIS AND ANALYSIS

Meta-analysis was performed using RevMan V.5.4 software and STATA 15.1 software. A range of outcome measurement data was extracted, including the mean, the number of animals (N), the standard deviation (SD) or the standard error of the mean (SEM). Whenever a unit of data was different, standardized mean difference (SMD) was applied without regard to whether it had been converted, otherwise the weighted mean difference (WMD) should be used. Heterogeneity among the included studies was evaluated using the $I$ square ($I^2$) statistical test, an analysis of random effects was conducted if $I^2$ was > 50%; otherwise, an analysis of fixed effects was conducted. The confidence interval (CI) was set at 95%, and a $p \leq 0.05$ was considered to be significant, publication bias was assessed by using funnel plots. Subgroup analysis were based on the dosage of emodin (<30 *vs.* ≥30 mg/kg/day), treatment duration (≤4 *vs.* >4 weeks), animal species (rat *vs.* mice).

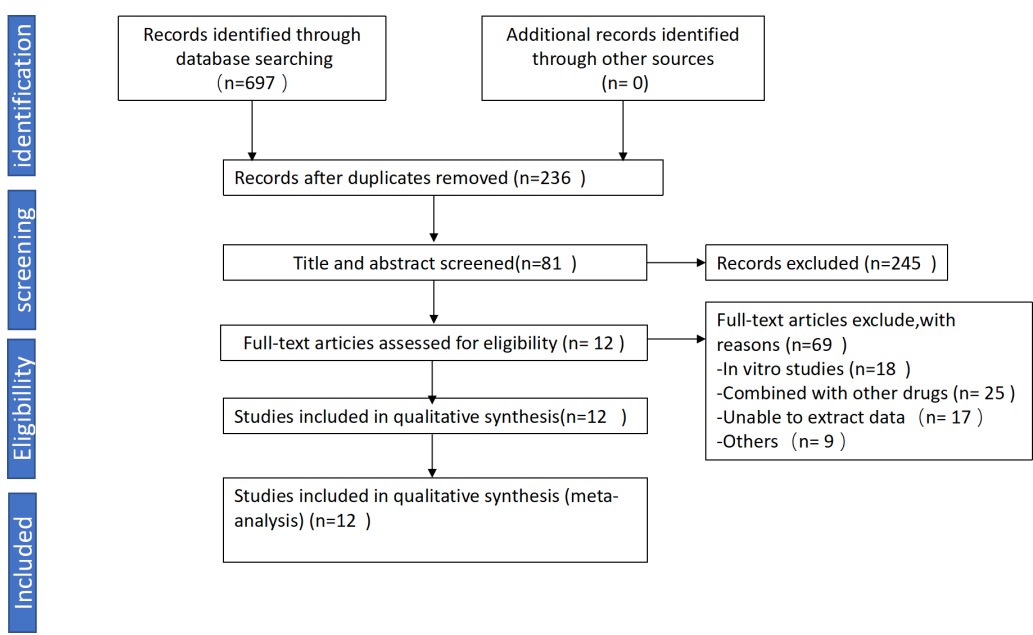

**Figure 1** **Flowchart for selection of studies.**

Two authors (Yang Xiao and Zhixuan Zhao) assessed the meta-analysis results using the GRADE evidence hierarchy system (https://gdt.gradepro.org/).

# RESULTS

## Study selection and characteristics of included studies

We collected a total of 697 articles, including 103 from PubMed, 151 from Embase, 67 from Web of Science, 81 from Cochrane Library, 85 from CNKI, 123 from VIP, 41 from Wanfang, 46 from CBM. After preliminary screening based on the aforementioned criteria, we excluded 92 unrelated studies and 369 duplicate studies. 245 articles were excluded due to lack of inclusion criteria. Ultimately, 12 studies involving 215 animals (110 in the experimental group and 105 in the control group) were included in this study (*Abu Eid et al., 2017*; *Arvindekar et al., 2015*; *Gao, 2019*; *Song & Liu, 2012*; *Song et al., 2017*; *Wang et al., 2012*; *Xiang et al., 2014*; *Xiao et al., 2019*; *Xue, Ding & Liu, 2010*; *Xuezheng et al., 2018*; *Zhao et al., 2009*; *Zhou et al., 2012*). The detailed search process is illustrated in Fig. 1. Among the 12 studies, four utilized male Wistar rats, two used C57BL/6 mice, three employed KKAy mice, one used ob/ob mice, one used SD male rats, and one used db/db mice.

## Quality of the included studies

Among the 12 included studies, only two mentioned baseline characteristics, three reported random housing conditions, and 11 studies provided complete outcome data. Most studies had a high risk of bias, mainly due to the lack of information on randomization, blinding, and sequence generation. The risk of bias assessment scores for all included studies were

**Table 1  Description of the characteristics of studies included in the systematic review.**

| Study | Species (strain; sex; age; weight) | Modeling method (establish; standard) | Anesthetic | Administration route (duration) | Treatment | Control | Outcome index | Intergroup differences |
|---|---|---|---|---|---|---|---|---|
| *Zhao et al. (2009)* | Rats (Wistar; M; NM; 200 ± 20 g) | By SIJ STZ (55 mg/kg); FBG ≥ 16.7 mmol/L | NM | By intragastric/ 2 weeks | Emodin, 80 mg/kg | Gavage of an equal volume of Saline | FBG, TC, TG, FFA | $P < 0.05$ |
| *Xue, Ding & Liu (2010)* | Mice (C57/BL6J; M; 8 week; NM) | By SIJ STZ (120 mg/kg); FBG ≥ 11 mmol/L | NM | By intraperitoneal/ 3 weeks | Emodin, 1.5 mg/kg | Gavage of an equal volume of Saline | FBG, IPGTT, TC, TG, HDL-C, LDL-C | $P < 0.05$ |
| *Wang et al. (2012)* | Mice (ob/ob mice; NM; NM; NM) | Spontaneously diabetic mutant | iv sodium pentobarbital (40 mg/kg) | By intraperitoneal/ 26 days | Emodin, 50 mg/kg | Gavage of an equal volume of Saline | FBG, OGTT, TC, TG, FFA | $P < 0.05$ |
| *Arvindekar et al. (2015)* | Rats (Wistar; M; NM; 180–200 g) | By SIJ STZ (70 mg/kg); FBG ≥ 200 mg/dl | NM | By intragastric/NM | Emodin, 3 mg/kg | Gavage of an equal volume of Saline | OGTT | $P < 0.05$ |
| *Xuezheng et al. (2018)* | Mice (Kkay; M; NM; NM) | Spontaneously diabetic mutant | sodium pentobarbital anesthesia (NM) | By intragastric/ 8 weeks | Emodin, 50 mg/kg | Gavage of an equal volume of Saline | FBG, INS, TC, TG, HDL-C, LDL-C, FFA | $P < 0.05$ |
| *Song et al. (2017)* | Mice (Kkay; M; 28 week; NM) | Spontaneously diabetic mutant | cervical dislocation | By intragastric/ 6 weeks | Emodin, 50 mg/kg | Gavage of an equal volume of Saline | FBG, INS, IPGTT | $P < 0.05$ |
| *Abu et al. (2017)* | Mice (C57/ BL6J; M; 6 week; NM) | NM | inhalation anaesthetic (Sevoflurane, Sigma) | By intragastric/ 8 weeks | Emodin, 5 mg/kg | Gavage of an equal volume of Saline | FBG, INS, OGTT | $P < 0.05$ |
| *Xiao et al. (2019)* | Rats (Wistar; M; 12 week; 180–220 g) | By SIJ STZ (25 mg/kg); FBG ≥ 16.7 mmol/L | NM | By intragastric/ 4 weeks | Emodin, 80 mg/kg | Gavage of an equal volume of Saline | FBG, TC, TG | $P < 0.05$ |
| *Xiang et al. (2014)* | Rats (Wistar; M; 12 week; 180–220 g) | By SIJ STZ (25 mg/kg); FBG ≥ 16.7 mmol/L | iv sodium pentobarbital (0.1 mL/kg) | By intragastric/ 4 weeks | Emodin, 100 mg/kg | Gavage of an equal volume of Sodium carboxymethyl cellulose solution | FBG, INS | $P < 0.05$ |
| *Song & Liu (2012)* | Mice (Kkay; M; 8week; 33–37 g) | Spontaneously diabetic mutant | NM | By intragastric/ 8 weeks | Emodin, 50 mg/kg | Gavage of an equal volume of Saline | FBG, INS, TC, TG | $P < 0.05$ |
| *Gao (2019)* | Rats (SD; M; 7 week; 180–220 g) | By SIJ STZ (60 mg/kg); FBG ≥ 16.7 mmol/L | NM | By intragastric/ 8 weeks | Emodin, 100 mg/kg | Gavage of an equal volume of Saline | FBG, OGTT | $P < 0.05$ |
| *Zhou et al. (2012)* | Mice (db/db; M; 8 week; NM) | Spontaneously diabetic mutant | NM | By intragastric/ 12 weeks | Emodin, 25 mg/kg | Gavage of an equal volume of Saline | FBG, TC, TG, Body weight | $P < 0.05$ |

**Notes.**

NM, Not mentioned; FBG, Fasting blood glucose; IPGTT/OGTT, Intraperitoneal Glucose Tolerance Test/ Oral Glucose Tolerance Test; TC, Total Cholesterol; TG, Triglycerides; LDL-C, Low density lipoprotein-cholesterol; HDL-C, High density lipoprotein-cholesterol; INS, Insulin; FFA, Free Fatty Acid.

from one to three, with one study scoring one, six studies scoring two, and five studies scoring three. Detailed results of the risk of bias assessment are presented in File S4.

## The results of outcome measures for meta-analysis

Regarding the study results, FBG were reported in 12 studies, six studies assessed 2hPG (IPGTT/OGTT), two studies assessed IPITT, seven studies reported levels of TG and TC, five studies provided data on serum insulin levels, and four studies recorded changes in weight. The basic characteristics of the included literature are shown in Table 1.

# OUTCOME MEASURES

## Primary outcome

### Glucose metabolism outcome

Among the 12 included studies, a random effects model was selected for FBG and 2hPG (IPGTT/OGTT) due to their high heterogeneity. The results indicated that emodin significantly reduced FBG (SMD = −4.79 (−6.47, −3.10), $P < 0.01$), suggesting that emodin positively impacts fasting blood glucose indicators in T2DM animal models. The Egger's test outcomes revealed ($t = −6.22$, $P = 0.00$). Coupled with the findings from the funnel plot, it suggested the presence of potential publication bias (Fig. 2). Compared to the

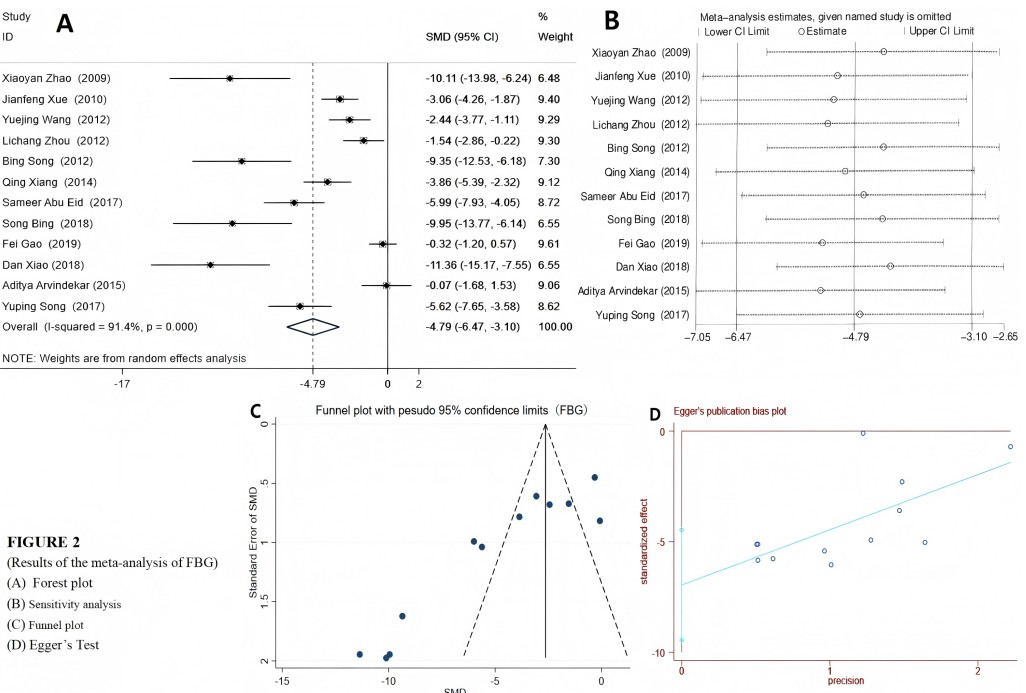

**Figure 2   (A–D) Forest plots of FBG.** Group treated with emodin; control: DM animal group. Studies: *Abu Eid et al., 2017*; *Arvindekar et al., 2015*; *Gao, 2019*; *Song & Liu, 2012*; *Song et al., 2017*; *Wang et al., 2012*; *Xiang et al., 2014*; *Xiao et al., 2019*; *Xue, Ding & Liu, 2010*; *Xuezheng et al., 2018*; *Zhao et al., 2009*; *Zhou et al., 2012*.

control group, T2DM animals treated with emodin exhibited a reduction in 2hPG (SMD = −2.85 (−4.22, −1.48), $P < 0.01$). It can be concluded that there is no publication bias in the results of this outcome indicator through the funnel plot and Egger's test ($P > 0.05$), as shown in Fig. 3. Since only two studies use IPITT as the outcome measures, the 2hPG in IPITT is described as follows: According to the studies by *Song et al. (2017)* and *Xue, Ding & Liu (2010)*, the 2hPG levels in the emodin-treated group were significantly lower than those in the control group ($P < 0.05$).

### Lipid metabolism index

We used a random-effects model to integrate the effect sizes of TC and TG. The results showed that the levels of TC and TG in the experimental group were significantly reduced: TC (SMD = −5.79 (−8.16, −3.43), $P < 0.001$) and TG (SMD = −4.84 (−5.89, −3.24), $P < 0.001$) (see Figs. 4 and 5). Only two studies provided data on LDL-c and HDL-c (*Xue, Ding & Liu, 2010*; *Xuezheng et al., 2018*), and the results indicated that after emodin treatment, the LDL-c level in the experimental group was significantly lower than that in the control group ($P < 0.001$), while the HDL-c level was significantly higher than that in the control group. The specific data are as follows:

- *Xuezheng et al. (2018)*: HDL-c in the experimental group: 2.89 ± 0.15 mmol/L, HDL-c in the control group: 1.79 ± 0.15 mmol/L.
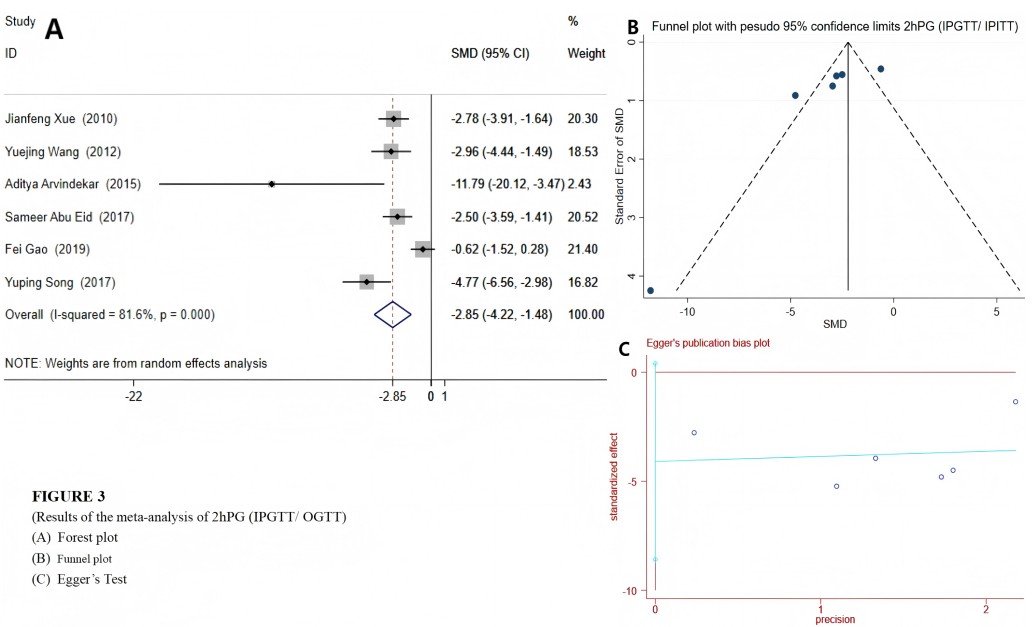

**Figure 3    Forest plots of 2hPG(OGTT/IPGTT).** Experimental: group treated with emodin; control: DM animal group. Studies: *Abu Eid et al., 2017*; *Arvindekar et al., 2015*; *Xue, Ding & Liu, 2010*; *Gao, 2019*; *Song et al., 2017*.

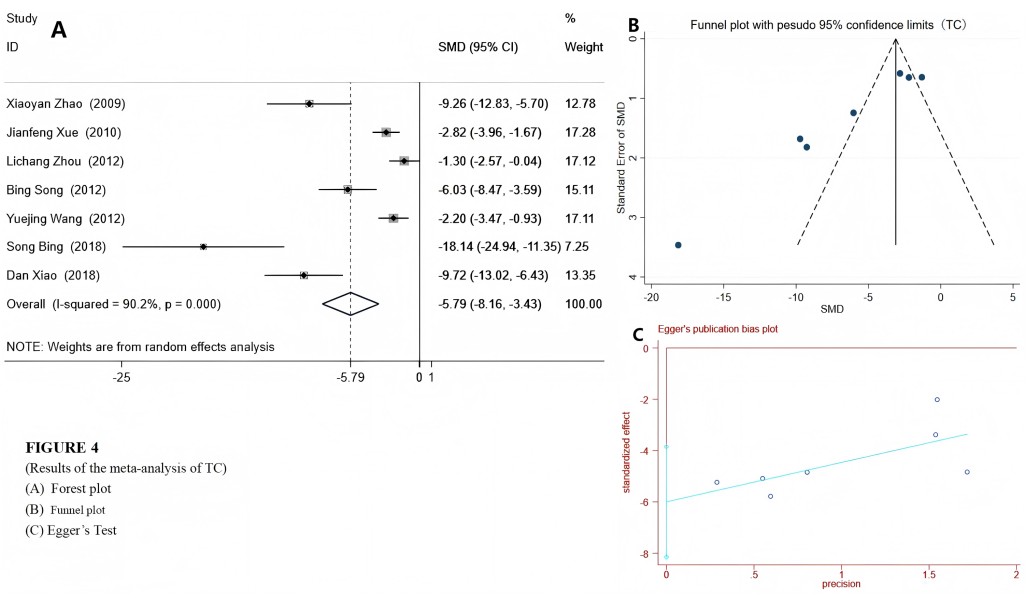

**Figure 4    Forest plots of TC.** Experimental group treated with emodin; control: DM animal group. Studies: *Zhao et al., 2009*; *Xue, Ding & Liu, 2010*; *Song & Liu, 2012*; *Wang et al., 2012*; *Song et al., 2017*; *Xiao et al., 2019*.

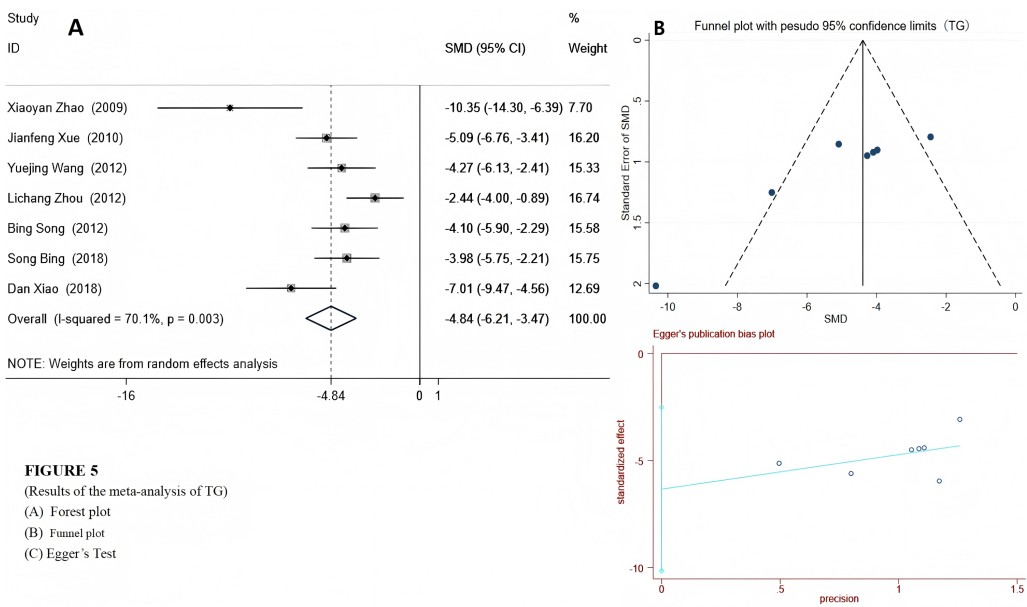

**FIGURE 5**
(Results of the meta-analysis of TG)
(A) Forest plot
(B) Funnel plot
(C) Egger's Test

**Figure 5 Forest plots of TG.** Experimental group treated with emodin; control: DM animal group. Studies: *Zhao et al., 2009*; *Xue, Ding & Liu, 2010*; *Wang et al., 2012*; *Zhou et al., 2012*; *Song & Liu, 2012*; *Song et al., 2017*; *Xiao et al., 2019*.

- *Xue, Ding & Liu (2010)*: HDL-c in the experimental group: $1.17 \pm 0.17$ mmol/L, HDL-c in the control group: $1.00 \pm 0.12$ mmol/L.

## Secondary outcome
### Serum insulin
The serum insulin levels of T2DM animals were also affected by emodin in five other studies. We used random-effects model to combine the effect sizes. The results indicated that the INS level in the experimental group was lower than that in the control group (WMD $= -24.43 \, (-33.94, -14.91)$, $P < 0.001$), as shown in File S5. Egger's test ($P = 0.812 > 0.05$) indicated no publication bias.

### Body weight
Four studies reported changes in body weight. Among them, *Xiao et al. (2019)* employed rats as the experimental subjects, while the other three studies used mice. As *Wang et al. (2012)* only reported the magnitude of body weight change without providing the baseline data of the original body weight, we only conducted a descriptive analysis of the results of Wang Yuejing's study, that is emodin can effectively reduce the weight of mice. The results of remaining three studies showed that the body weight loss in the experimental group was greater than control group (WMD $= -18.66 \, (35.82, -1.51)$), the relevant results are shown in File S6. Among them, one study reported an increase in body weight following treatment (*Xue, Ding & Liu, 2010*), while other two studies indicated that emodin can reduce body weight (*Xiao et al., 2019*; *Zhou et al., 2012*). The Egger's test results indicated no publication bias ($P = 0.562 > 0.05$).

**Table 2  The results of outcome measures for meta-analysis.**

| Outcome measures | Studies | Quantity of studies | N(E/M) | Heterogeneity | | Meta-analysis | | Test of SMD/WMD | | Egger' test | |
|---|---|---|---|---|---|---|---|---|---|---|---|
| | | | | $I^2\%$ | $p$ | SMD/WMD | 95% CI | $Z$ | $p$ | $t$ | $p$ |
| FBG | Arvindekar et al. (2015), Gao (2019), Song et al. (2017), Song & Liu (2012), Wang et al. (2012), Xiang et al. (2014), Xiao et al. (2019), Xue, Ding & Liu (2010), Xuezheng et al. (2018), Zhao et al. (2009), Zhou et al. (2012), Abu et al. (2017) | 12 | 215 (110/105) | 91.4 | <0.001 | −4.79 | −6.47, −3.10 | 5.58 | <0.01 | −6.22 | 0.000 |
| 2hPG | Arvindekar et al. (2015), Gao (2019), Song et al. (2017), Wang et al. (2012), Xue, Ding & Liu (2010), Abu et al. (2017) | 6 | 111 (58/53) | 81.6 | <0.001 | −2.85 | −4.22, −1.48 | 4.07 | <0.01 | −2.53 | 0.065 |
| TC | Song & Liu (2012), Wang et al. (2012), Xiao et al. (2019), Xue, Ding & Liu (2010), Xuezheng et al. (2018), Zhou et al. (2012) | 7 | 121 (63/58) | 90.2 | <0.001 | −5.97 | −8.16 , −3.43 | 4.80 | <0.01 | −7.17 | 0.001 |
| TG | Song & Liu (2012), Wang et al. (2012), Xiao et al. (2019), Xue, Ding & Liu (2010), Xuezheng et al. (2018), Zhou et al. (2012) | 7 | 121 (63/58) | 70.1 | <0.001 | −4.84 | −6.21, −3.47 | 6.91 | <0.01 | −4.26 | 0.008 |
| INS* | Song et al. (2017), Song & Liu (2012), Xiang et al. (2014), Xuezheng et al. (2018), Abu et al. (2017) | 5 | 92 (46/46) | 96.5 | <0.001 | −12.63 | −21.03, −4.24 | 2.95 | =0.03 | −0.27 | 0.812 |
| Weight* | Xiao et al. (2019), Xue, Ding & Liu (2010), Zhou et al. (2012) | 3 | 57 (31/26) | 97.2 | <0.001 | −18.66 | −35.82, −1.51 | 2.13 | <0.05 | −5.15 | 0.122 |

**Notes.**

N, Number; E, Emodin group; M, Model group; SMD, Standardized Mean Difference; WMD, Weighted Mean Difference.

*Statistical analysis using WMD.

## META REGRESSION AND SUBGROUP ANALYSIS

In this study, as there was significant heterogeneity in the outcome measures (Table 2), we employed univariate meta-regression to identify the sources of heterogeneity (respectively for the three factors of drug dose, animal species, and intervention duration). The boundaries for the relevant factors were set as follows: emodin dosage <30 mg/kg and ≥30 mg/kg, intervention duration ≤4 weeks and >4 weeks, and animal species are categorized as either rats or mice. We carried out meta-regression for FBG, 2hPG (IPGTT/OGTT), TG, TC, INS, and weight respectively. If the result of the univariate regression is statistically significant ($P < 0.05$), we further conducted subgroup analysis. The relevant results of meta regression are shown in File S7.

The results of the meta-regression indicate that that emodin dosage might be a factor contributing to the high heterogeneity of INS ($P < 0.05$), while animal species was a significant cause of the high heterogeneity of TG ($P < 0.05$). Therefore, we conducted a subgroup analysis on INS and TG.

After subgroup analysis of TG based on different animal species, the heterogeneity was significantly reduced. The results showed that emodin significantly improved the TG indicator in both rats and mice (rats: $I^2 = 49.3\%$, $P = 0.160$, SMD $= -8.30$ [−11.49, −5.12]; mice: $I^2 = 27.1\%$, $P = 0.241$, SMD $= -3.93$ [−4.83, −3.02]). However, after subgroup analysis, the heterogeneity of INS didn't decrease. The results are presented in File S8.

## SENSITIVITY ANALYSIS

If the number of included articles exceeds 10, we will conduct a sensitivity analysis on the corresponding outcome indicators. In this study, 12 articles reported FBG, so we conducted a sensitivity analysis, and the results were robust. We used RevMan to exclude each study one by one, and the heterogeneity results remained unchanged. Additionally, we also conducted sensitivity analyses on other outcome measures (2hPG, TG, TC, INS, and body weight), and the results were equally stable (the sensitivity analysis results for less than 10 studies are presented in Files S9–S13).

## PUBLICATION BIAS

This study evaluated publication bias by drawing inverted funnel plots of FBG, 2hPG (IPGTT/OGTT), TC, TG, INS and body weight. The results showed that the funnel plots of FBG, TG and TC were asymmetrical, suggesting the presence of publication bias (Egger's test, $P < 0.05$). Therefore, we applied the trim and fill method for correction, but no studies needed to be filled in were found (see Files S14–S16). After deleting the studies with poor symmetry, publication bias disappeared, but heterogeneity remained significant, and it does not affect our final conclusion, indicating that the impact of publication bias on the results of this study was limited. The reasons for this result may include high heterogeneity, a small number of included studies, and inconsistent directions of publication bias. This result may change after more high-quality studies are included in the future.

## THE ASSESSMENT RESULTS OF THE GRADE EVIDENCE RATING SYSTEM

The present study, based on animal experiments, analyzes the efficacy of emodin. We conducted a GRADE assessment of evidence quality for five outcome indicators, which were rated as "very low", the specific results are presented in File S17.

Regarding bias risk, only two original studies explicitly reported the baseline consistency between experimental and control groups. Most studies did not clearly describe processes such as random sequence generation, allocation concealment, and blinding; thus, maybe there is a high risk of bias across various domains. Consequently, we considered downgrading each of these five outcome indicators by one level due to bias risk.

In terms of inconsistency, there are notable differences in the animal species and the interventions across studies. Meta-analysis revealed significant heterogeneity among relevant outcome indicators; however, after subgroup analysis for TG, both groups exhibited $I^2 \leq 50\%$. Therefore, with the exception of TG, all other five outcome indicators were downgraded by one level.

Regarding imprecision, although none of the 95% confidence intervals for all outcome indicators crossed the null line, except for FBG, the sample sizes for four out of five outcomes were small. Thus, we considered downgrading these indices accordingly.

Concerning indirectness, this study included various rat or mouse models that represent lower organisms; hence a downgrade was warranted. Although the doses and durations

of the interventions and controls in different studies varied, all these interventions were within the normal range and all involved emodin. Therefore, we do not believe that this indirectness had an impact.

Finally, regarding publication bias: Egger's test indicated that there was potential publication bias for FBG, TC, and TG, we considered downgrading each of these three outcome measures by one level; however, no significant publication bias was observed in the results of 2hPG, INS, or body weight, so the evidence level for these was not adjusted.

## DISCUSSION

Emodin is the primary active component of the Chinese rhubarb, which has a long history of application in China. Chinese rhubarb possesses properties that promote blood circulation and alleviate stasis, as well as facilitate bowel movements. Therefore, its clinical usage requires caution, for patients with weak, it is generally advisable to reduce the dosage. In recent years, emodin has been proven to be able to improve disorders of glucose and lipid metabolism. Preliminarily, we investigated emodin's efficacy and possible mechanism of action against T2DM in diabetic animal models using preclinical systematic evaluation. A total of 215 animals were included, 110 of which were treated with emodin. Emodin exerted significant anti-T2DM effects in animal models, mainly manifested in the following aspects: decreasing FBG, 2hPG (IPGTT/OGTT) and IPITT ($P < 0.001$), reducing TG, TC and LDL-c ($P < 0.001$), lowering INS ($P < 0.001$), and improving the index of HDL-c ($P < 0.001$). Furthermore, we observed its potential for body weight reduction and potentially enhancing insulin sensitivity. The improvement in insulin resistance is reflected in the decrease of serum insulin levels and the reduction in body weight. Subsequently, we conducted a GRADE assessment of the evidence levels, revealing that all outcome indicators were rated as very low. Furthermore, significant heterogeneity was observed among the studies. Therefore, it is essential to approach the results with caution.

### Emodin exhibits remarkable efficacy in improving blood glucose levels

Regarding blood glucose levels, emodin has demonstrated remarkable hypoglycemic effects. In the 12 studies included, both low and high doses of emodin consistently reported a significant positive impact on improving fasting blood glucose (FBG). Notably, in T2DM models induced by genetically deficient mice such as db/db and ob/ob mice, emodin also exhibited an ability to improve FBG levels. Similar effects were observed in the 2hPG (IPGTT/ OGTT) or IPITT tests, which further substantiates the hypoglycemic properties of emodin.

### Emodin can improve lipid metabolism disorder

In terms of blood lipids, our findings in 9.2 indicate that emodin can regulate lipid metabolism disorders. Emodin has a certain effect in reducing LDL-c levels and increasing HDL-c levels. From the previous meta-regression and subgroup analyses, we observed that emodin can reduce TC and TG levels. However, after conducting a meta-analysis on these two outcome indicators, we found high heterogeneity. Although we could not identify the

source of heterogeneity for TC, we found that animal species might be the cause of the high heterogeneity in TG.

Following subgroup analysis based on species, the results showed that the heterogeneity of both subgroups was significantly reduced. It is worth noting that among the studies on TG result measurement that we analyzed, only two articles used rats as research subjects, while the other five chose mice. Among them, a 12-week study (*Xiao et al., 2019*) reported that Wistar rats, after being treated with 80 mg/kg of emodin, the TG level in the treatment group was $18.27 \pm 0.98$ mmol/L (mean $\pm$ SD), while that in the control group was $38.21 \pm 3.9$ mmol/L (mean $\pm$ SD); this left us puzzled, and we were unsure if any instrument or reagent could accurately detect such a high upper limit, or if there were errors in these data or units. After in-depth discussion, we concluded that even excluding this study would not change the fact that emodin effectively reduces TG levels in mice; for the efficacy of emodin in lowering TG levels in rats, we remain cautious. As more high-quality studies on the impact of emodin on TG levels in rats emerge, the evidence supporting emodin's ability to decrease TG levels in rats will become increasingly compelling.

### Emodin could reduce body weight in animal models of T2DM

Four studies reported changes in body weight. We only conducted a meta-analysis on three of the studies. The meta-analysis results indicated that the weight loss in the experimental group was greater than that in the control group (WMD $= -18.66\,(35.82, -1.51)$). Among these, one study observed an increase in mice body weight (WMD $= 2.20\,(-0.71, 5.11)$) (*Xue, Ding & Liu, 2010*), the research used 8-week-old male C57BL/6 mice and lasted for 21 days with a dosage of only 1.5 mg/kg. From the 6th day after the intervention with emodin, the inhibitory effect of STZ on body weight gradually decreased, but there was no statistically significant difference in body weight between the treatment group and the model group. In contrast, the remaining two studies recorded weight loss in the animal models. The two studies employed genetically modified mice: db/db mice and ob/ob mice respectively (*Wang et al., 2012*; *Zhou et al., 2012*), both commonly used animal models in diabetes research based on leptin or leptin receptor gene modifications. Specifically, db/db mice exhibit normal leptin synthesis but have defective leptin receptors due to mutations located on chromosome 4; conversely, ob/ob mice have impaired leptin synthesis while their receptors remain functional due to mutations on chromosome 6. Consequently, both types typically present with increased body weights and obesity characteristics (*Suriano et al., 2021*). After treatment with emodin, the body weights of both types of transgenic mice decreased. However, it is important to note that without intervention, these genetically deficient mice begin gaining weight at ages ranging from 3 to 8 weeks; by 3 to 6 months old their insulin levels start declining alongside subsequent decreases in body weight (*Keller et al., 2009*). Therefore, this objective observation indicates that the weight loss in the gene-deficient mice might be the result of the combined effect of their inherent physiological characteristics and the therapeutic action of emodin. Another study utilized Wistar rats as subjects for inducing T2DM models through STZ combined with high-fat diets (*Xiao et al., 2019*). Although STZ injections initially led to some reduction in rat weights after

administration, over time their weights began increasing again; data presented within this article indicate that emodin can indeed lower rats body mass.

In addition, among the four studies, three employed doses of emodin that were significantly higher than that used in the study by *Xue, Ding & Liu (2010)*. The doses administered were 25 mg/kg, 50 mg/kg, and 80 mg/kg, whereas *Xue, Ding & Liu (2010)* utilized a dose of only 1.5 mg/kg. This dosage is considerably lower than those applied in two other studies involving mice (*Wang et al., 2012*; *Zhou et al., 2012*). Therefore, aside from the weight suppression caused by STZ and the inherent physiological characteristics of the genetically modified mice, one possible reason for the differing results between these three studies and that of *Xue, Ding & Liu (2010)* may be attributed to this substantial discrepancy in emodin dosages.

The aforementioned findings are based on results derived from animal models. Furthermore, Chinese rhubarb has been employed within traditional Chinese medicine for thousands of years where practitioners have accumulated extensive experience regarding its use: Chinese rhubarb possesses potent laxative effect; prolonged high-dose usage may deplete vital energy leading individuals towards weakness while adversely affecting gastric function which can influence appetite and digestion thereby impacting changes in overall body mass.

In summary, after thorough discussions within our team, we continue to affirm the findings of this study, which indicate that emodin can reduce body weight in T2DM models. However, it is noted that the impact of low doses of rhubarb extract on body weight is minimal.

## Emodin release a positively signaled in insulin resistance

Five studies reported the insulin levels. The results indicate that emodin can significantly reduce serum insulin levels (WMD = $-24.43$ ($-33.94$, $-14.91$), $P < 0.001$). Excessive secretion of insulin inhibits the breakdown of fats, leading to weight gain. This increase in body weight further exacerbates insulin resistance, diminishing peripheral tissue sensitivity to insulin and impairing the effective utilization and metabolism of glucose. Notably, emodin demonstrates significant improvements in blood sugar regulation, weight reduction, and lowering serum insulin levels within this research context which will contribute to the improvement of insulin resistance.

## Possible mechanisms

To mitigate the prevalence of T2DM and offer improved treatment for patients, it is imperative for us to delve deeper into the mechanism of action of emodin in treating T2DM (*Heng et al., 2020*; *Khatami, Mohajeri-Tehrani & Tavangar, 2019*), identify common targets of action between emodin and other effective medications, thereby broadening the treatment options for T2DM. The pathogenesis of T2DM can be attributed to eight factors. In addition to the traditional triad of factors—insulin secretion deficiency, reduced glucose uptake by muscle tissue, and increased glucose output by the liver, many organs and hormones are also involved in the occurrence and development of T2DM. These factors include lipid metabolism disorders, weakened incretin effect, elevated basal glucagon levels,

increased renal glucose reabsorption, and neurotransmitter dysfunction in the brain. These complex pathophysiological mechanisms collectively form the "octet" of the pathogenesis of T2DM (*Defronzo, 2009*). In this study, we discovered that emodin can improve blood glucose levels *via* the aforementioned macroscopic mechanisms. Here is a brief summary of these mechanisms:

(1) Enhance glucose uptake in peripheral tissues. Emodin enhances liver glucose utilization, glucose uptake in muscle and fat *via* the IRS/PI3K/Akt/FoxO1 pathway, similar to the mechanism of action of metformin. The phosphoinositide 3-kinase(PI3K)/protein kinase B (Akt) signaling pathway is a crucial insulin signaling pathway, playing a significant role in numerous inflammatory mechanisms. Diabetes is also considered an inflammatory response, with the PI3K/Akt pathway being involved in this reaction. FoxO1, a transcription factor negatively regulated by insulin signaling, is the first downstream target protein confirmed to be regulated by Akt (*Behl et al., 2022*). Emodin can downregulate FoxO1 expression through this pathway, thereby inhibiting liver glycogen output (*Ganesan & Xu, 2019*; *Qi et al., 2022*; *Walkowski et al., 2022*). Additionally, emodin can increase GluT-4 expression. GluT-4 protein is a key protein downstream of the PI3K/Akt pathway, related to glucose transport, primarily located on the cell membrane. After Akt phosphorylation, it accelerates the transfer of GluT-4 from cells to the cell membrane, fuses with the cell membrane, and increases glucose uptake (*Herman et al., 2022*; *Koepsell, 2020*).

(2) Inhibit the activity of alpha-glucosidase (AG). Emodin exhibited potent intestinal α-glucosidase (AG) inhibitory activity, with a significantly lower IC50 compared to acarbose, which is similar to the mechanism of action of acarbose (*Arvindekar et al., 2015*).

(3) Improve insulin resistance. The first point mentioned is that emodin can enhance the glucose uptake by peripheral tissues, thereby improving insulin resistance to a certain extent. Additionally, studies have found that miR-20b is a crucial molecule regulating metabolism and inflammation, and it is prevalent in type 2 diabetes (*Villard et al., 2015*; *Ye et al., 2018*). Emodin downregulates miR-20b and upregulates SMAD7, thereby improving glucose metabolism and exerting anti-IR effects.

(4) Enhance sensitivity to insulin. 11β-HSD1 can enhance the activity of glucocorticoids and amplify their effects, directly influencing insulin sensitivity. Therefore, 11β-HSD inhibitors may emerge as new targets for treating diabetes, obesity, and other diseases. Emodin can precisely inhibit the activity of this new target, 11β-HSD1, thereby regulating blood glucose levels (*Hollis & Huber, 2011*; *Pereira et al., 2012*; *Yao et al., 2017*).

(5) Increase the expression of L-type calcium channels. Insulin secretion is triggered by the influx of $Ca^{2+}$ through voltage-dependent L-type calcium channels (*Mears, 2004*). The expression of Cav1.2 channels in the pancreas of diabetic rats is significantly downregulated, and there was a varying degree of recovery after emodin treatment. These data suggest that emodin can improve the expression of Cav1.2 in the pancreas, which may be one of the mechanisms underlying emodin's anti-diabetic effects (*Zhao et al., 2009*).

(6) Emodin promotes insulin secretion, stimulates the secretion of incretin and protects the function of pancreatic islet cells. It facilitates glucose metabolism by increasing GLP-1 expression. GLP-1, a peptide hormone secreted by the intestine, acts on pancreatic beta cells to promote insulin synthesis and secretion, emodin stimulate their proliferation and

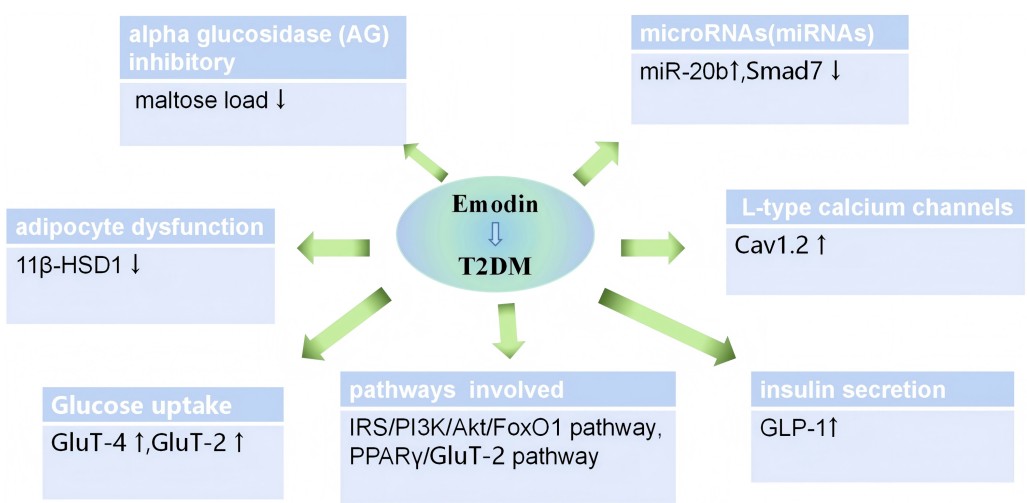

**Figure 6** Mechanisms of emodin for glucose and lipid metabolism disorder on diabetic animals.

differentiation, inhibit apoptosis, and increase their number, thereby protecting pancreatic function (*Razavi et al., 2022*; *Zhu et al., 2022*).

(7) Regulate lipid metabolism balance. PPARγ is an important nuclear receptor factor regulated by ligands in the body. Pathological changes and insulin resistance occur following the loss of PPARγ function or gene mutations, leading to the development of T2DM. The activation of PPARγ may be involved in the anti-diabetic effect of emodin. Emodin activates PPARγ, thereby regulating lipid metabolism and improving blood glucose levels (*Frkic, Richter & Bruning, 2021*; *Taghvaei & Saremi, 2022*).

Emodin is used to treat T2DM through various mechanisms. Apart from its effects similar to SGLT-2 inhibitors, which are currently unconfirmed, and its ability to regulate neurotransmitter pathways, emodin also addresses the other six causes of T2DM.

At the microscopic level of molecular biology, the effects of emodin can be summarized as anti-inflammatory mechanisms, immune-inflammatory crosstalk mechanisms, and the production of IFN by beta cells mediated through calcium-mediated glycolysis. We have summarized the main confirmed mechanisms, as shown in Fig. 6. In addition, several studies indicate that the efficacy of emodin is linked to its ability to inhibit the activation of various protein kinases involved in oxidative stress and inflammatory processes (*Cui et al., 2020*).

## Suggestions, precautions and future prospects for the medicinal application of emodin

Among the 12 included studies, no reports of liver or kidney function damage were found. Research shows that emodin has a hypoglycemic effect, but no hypoglycemic phenomenon was observed in animals, indicating that emodin is relatively safe within a certain dose range. A 28-day toxicity study indicated that low and medium doses (100 and 300 mg/kg) of emodin did not cause liver damage in mice, while a dose of 600 mg/kg led to obvious hepatotoxicity (*Huo et al., 2021*). One study also pointed out

that excessive emodin (1 g/kg) could cause liver damage in SD rats, the mechanism might be due to a significant increase in the production of primary bile acids, which affected the expression of FXR-RXR-CYP7A1 and inhibited the expression of NTCP in the liver, thereby disrupting the enterohepatic circulation of bile acids andultimately inducing liver damage (*Tian et al., 2025*). A study shows that intragastric administration of 80 mg/kg can effectively improve the disorder of glucose and lipid metabolism in C57BL/6 mice, suppress their food intake, reduce body weight, and enhance insulin sensitivity. However, the dose of 40 mg/kg does not affect food intake, TC, TG levels or insulin sensitivity (*Li et al., 2016*). One study has shown that intraperitoneal injection of 10 mg/kg of emodin can down-regulate the expression of NF-κB, thereby exerting a protective effect on liver function in rats with acute liver injury (*Yu & Wang, 2017*), low-dose and short-term administration may have a liver-protective effect, high doses may increase the risk of inducing liver damage (*Tian et al., 2025*). Therefore, high doses and long-term use are important factors contributing to the toxic and side effects of emodin (*Liu, Song & Sun, 2024*), which are particularly important in human clinical trials and deserve special attention. Additionally, the study mentioned that during the extraction of total anthraquinones from rhubarb, after thermal treatment and acid hydrolysis, anthraquinone glycosides are hydrolyzed into aglycones. By using oral colon-targeting technology to directly release the aglycones into the colon, the hepatotoxicity of anthraquinone drugs from rhubarb can be significantly reduced (*Liu et al., 2009*). One study demonstrated that through processing, the tannin and bound anthraquinone content in rhubarb can be reduced, while the content of free anthraquinones can be increased, thereby significantly reducing hepatotoxicity and gastrointestinal irritation (*Wang et al., 2009*). In summary, by optimizing the processing techniques and extraction methods of rhubarb, the efficacy of emodin can be enhanced while reducing toxicity, providing a safer and more effective approach for clinical application.

Given that the majority of the administration routes included in this study are oral gavage and that emodin has a significant improvement effect on glucose and lipid metabolism through this route, we suggest that the clinical administration route of emodin be oral. The initial dose should start from a small dose and be gradually adjusted according to individual differences. The specific dosage needs to be further precisely determined through human pharmacodynamics, pharmacology and toxicology studies to establish the medication standards. In addition, future research should focus on improving the oral bioavailability of anthraquinones from rhubarb and reducing their toxicity to expand their clinical application prospects.

At present, many basic hypoglycemic drugs such as metformin and acarbose all have certain adverse reactions in the gastrointestinal tract. This provides potential application prospects for emodin in the treatment of disorders of glucose and lipid metabolism. For patients who cannot tolerate metformin, under the premise of ensuring efficacy, the dose of metformin can be reduced and combined with a small dose of emodin for treatment. Similarly, acarbose, as a first-line drug for controlling postprandial blood glucose, is often accompanied by adverse reactions such as abdominal distension and diarrhea, especially for patients with a history of intestinal obstruction, it needs to be used with caution or even prohibited. This study shows that emodin has a certain effect in reducing postprandial blood

glucose, so it may become an effective natural extract drug for improving postprandial blood glucose, and no adverse reactions in the gastrointestinal tract have been reported. In addition, since emodin also shows sensitization effects and can improve insulin resistance, it is also expected to become a new insulin sensitizer, providing more treatment options for patients.

### Some recommendations for future researchers

The results indicate that emodin can regulate blood lipids, blood glucose, and insulin levels in diabetic patients. Furthermore, when evaluating the therapeutic effect of emodin, its safety cannot be ignored (*Cui et al., 2020*). It is crucial to prevent potential harm to body health from prolonged high-dose usage, which may lead to patient weakness. In this context, we propose several recommendations:

(1) Record indicators such as the mental state, fur color, grip strength, and claw strength of rats to indirectly assess the safety of emodin.

(2) Given that emodin has a strong laxative effect, researchers should monitor stool morphology in T2DM animal models.

(3) Encouraging the conduct of pharmacodynamic studies on a single-dose treatment of emodin for T2DM which will provide evidence for subsequent clinical dosage applications.

(4) Liver and kidney functions in T2DM animal models should be assessed to furnish additional evidence regarding the safety of emodin extract treatment for T2DM (*Liu et al., 2022*).

(5) Emodin has been shown to aid in weight loss; since it is a primary component of Chinese rhubarb, prolonged use may impair gastric qi and affect appetite and digestion, so this raises questions about whether its weight loss and blood glucose-lowering effects are achieved at the expense of appetite and food intake in T2DM animal models or through direct reductions in blood glucose levels and insulin concentrations. Therefore, meticulous recording of food intake and body weight during experiments is necessary. These findings will serve as critical data for assessing the future safety profile and adverse reactions associated with bringing emodin as a new drug to market, all these aspects are vitally important considerations.

## LIMITATIONS

There are also some limitations to the present study. First of all, due to the fact that we searched all databases in English or Chinese, there might be a selection bias as studies published in other languages may be excluded. Second, studies included in this review have an average methodological quality. Study selection, information bias, and publication bias may be caused by the lack of blinded assessments and sample size calculations in the included studies. Third, since the units of included data were different, the data might have been measured differently. Therefore, the present study contains some conclusions that require critical evaluation.

## CONCLUSIONS

The results were treated with caution due to the significant heterogeneity among the studies. In conclusion, emodin demonstrates remarkable potential in the treatment of diabetes, it could effectively reducing FBG, 2hPG (IPGTT/ OGTT), IPITT, TC, TG, insulin levels and body weight in T2DM animal models. Nevertheless, whether the reduction of TG and body weight by emodin is affected by the species difference between rats and mice requires more high-quality experimental data to reach more persuasive conclusions. Constrained by multiple factors such as species, dosage, and the quantity of literature, no obvious association has been observed currently between the effect of emodin on serum insulin levels and the drug dosage. The main mechanism underlying the therapeutic efficacy of emodin in type 2 diabetes is being accomplished by enhancing the utilization of glucose in peripheral tissues, inhibiting the absorption of glucosidase, alleviating insulin resistance, and strengthening L-type calcium channels, among other pathways. Emodin not only promoting insulin secretion but also enhancing the cells' sensitivity to insulin. Furthermore, emodin also presents actions similar to those of GLP-1 receptor agonists, suggesting its potential for protecting target organs. Consequently, it is an extremely promising drug with significant research and clinical value.

**Abbreviations**

| | |
|---|---|
| **T2DM** | Type 2 diabetes mellitus |
| **SMD** | Standardized mean difference |
| **WMD** | Weighted mean difference |
| **IDF** | International Diabetes Federation |
| **CBM** | China Biology Medicine |
| **CNKI** | China National Knowledge Infrastructure |
| **MeSH** | Medical subject headings |
| **FBG** | Fasting blood glucose |
| **2hPG in the IPGTT** | 2-h plasma glucose in the intraperitoneal glucose tolerance test |
| **OGTT** | Oral glucose tolerance test |
| **2hPG in the IPITT** | 2-h plasma glucose in the intraperitoneal insulin tolerance test, |
| **TG** | Serum/plasma triglyceride |
| **TC** | Serum/plasma total cholesterol |
| **HDL-c** | High-density lipoprotein cholesterol |
| **LDL-c** | Low-density lipoprotein cholesterol |
| **GLP-1** | Glucagon-likepeptide-1 |

### Funding

This study was supported by the National Natural Science Foundation of China (grant number: 81973813); Shenzhen Science and Technology Innovation Program (grant number: JCYJ20230807120510022, JCYJ20240813160612016, JCY20190809110015528); Science and Technology Development Plan Project of Jilin Province, China (grant

number YDZJ202401684ZYTS); Education Department of Jilin Province (grant number JJKH20241053KJ, JJKH20241050KJ, JJKH20241063KJ); Futian Healthcare Research Project (NO. FTWS070). The funders had no role in study design, data collection and analysis, decision to publish, or preparation of the manuscript.

## Grant Disclosures

The following grant information was disclosed by the authors:
National Natural Science Foundation of China: 81973813.
Shenzhen Science and Technology Innovation Program: JCYJ20230807120510022, JCYJ20240813160612016, JCY20190809110015528.
Science and Technology Development Plan Project of Jilin Province, China: YDZJ202401684ZYTS.
Education Department of Jilin Province: JJKH20241053KJ, JJKH20241050KJ, JJKH20241063KJ.
Futian Healthcare Research Project: NO. FTWS070.

## Competing Interests

The authors declare there are no competing interests.

## Author Contributions

- Yang Xiao conceived and designed the experiments, performed the experiments, analyzed the data, authored or reviewed drafts of the article, and approved the final draft.
- Zhixuan Zhao conceived and designed the experiments, performed the experiments, analyzed the data, authored or reviewed drafts of the article, and approved the final draft.
- Binqin Chen analyzed the data, prepared figures and/or tables, authored or reviewed drafts of the article, and approved the final draft.
- Jian Sun analyzed the data, prepared figures and/or tables, and approved the final draft.
- Li Wang conceived and designed the experiments, prepared figures and/or tables, and approved the final draft.
- Yu Wang analyzed the data, prepared figures and/or tables, and approved the final draft.
- Zheng Nan conceived and designed the experiments, prepared figures and/or tables, authored or reviewed drafts of the article, and approved the final draft.
- Qi Zhang conceived and designed the experiments, performed the experiments, analyzed the data, authored or reviewed drafts of the article, and approved the final draft.

## Data Availability

   This is a systematic review/meta-analysis.

## Supplemental Information

Supplemental information for this article can be found online at http://dx.doi.org/10.7717/peerj.19221#supplemental-information.

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
