# Peer review of "Emodin, a rising star in the treatment of glycolipid metabolism disorders: a preclinical systematic review and meta-analysis"

_PeerJ, doi:10.7717/peerj.19221_

## Round 0.1 · original submission · Major Revisions

Kindly revise your, manuscript it needs extensive grammatical revision.
the discussion needs to include more studies about Emodin and its medicinal uses.
What about its toxicity?
Is it safe to be used medicinally?
What is the suggested route of administration?
Try to add a paragraph to the discussion section writing each of the previous points in detail.

Reviewer 1 ·

Basic reporting

The manuscript should maintain a consistently high level of professional English to ensure that the content is clear and unambiguous. I suggest a thorough linguistic revision, especially for the abstract and conclusion sections, as they are essential for understanding the study's impact and implications.
Although the background information provided is relevant, it could be further strengthened by including the most recent literature. This would offer a more complete context for the role of emodin in the current pharmacological landscape for the treatment of T2DM.

Experimental design

The research question is clearly articulated, but the manuscript could be strengthened by providing a more explicit discussion on how this systematic review addresses a distinct gap in the current understanding of T2DM treatments.
The utilization of SYRCLE’s risk of bias tool and statistical software for data analysis is a suitable choice. Nonetheless, the manuscript would benefit from a more detailed account of the selection criteria for including studies and the methodology employed for data extraction.

Validity of the findings

The conclusion that emodin significantly reduces glucose and lipid metabolism indicators is an exciting finding.However,it is essential that the manuscript confirms the statistical reliability of the data supporting these claims and that potential confounding factors have been adequately controlled for.
The conclusions presented should be directly connected to the original research question and should only include the results that have been obtained from the study.Additionally,any suggestions for future research should be explicitly identified as hypotheses or recommendations to distinguish them from the established findings.

Additional comments

The manuscript does recognize the considerable heterogeneity among the studies included in the meta-analysis. To strengthen the analysis, it is essential to delve deeper into this issue. One effective approach could be to perform subgroup analyses that take into account varying types of T2DM models or different dosages of emodin. This methodological refinement can provide a more nuanced understanding of how emodin affects glucose and lipid metabolism across diverse conditions.
In light of the potential publication bias, as suggested by the Egger’s test for certain outcome measures, the authors should thoroughly discuss the potential impact of this bias on their findings. Additionally, sensitivity analyses should be conducted to evaluate the robustness of the conclusions.

·

Basic reporting

In my opinion, the manuscript is well-organized and follows PRISMA guidelines, indicating transparency and adherence to established systematic review protocols. However, I have some comments listed below:

1. Please thoroughly check for grammatical errors that may hinder readability (e.g.in Abstract (Lines 36-37):“it could effectively reducing FPG…”), the same with Conclusion (Lines 470-472) "it could effectively reducing FPG...".
Another example, Methods (Lines 79-81) "We chose PRISMA to guide the entire system and review process(Page et al., 2021). and registered in detail on.."

2. Suggest to incorporate more recent studies (from 2020-2024) to reflect the latest advancements in the field.

3. Figures and tables lack detailed descriptions that would aid readers in understanding the data.

4. Although the authors mention raw data availability, no direct links to datasets are provided within the main text.

Experimental design

1. Perform additional subgroup analyses or sensitivity tests to better explain the sources of heterogeneity.

2. Please provide a rationale for including different animal models or doses of emodin and address potential limitations.

Validity of the findings

1. Funnel plots and Egger’s tests reveal potential publication bias, but this issue is not sufficiently addressed in the discussion.

2. The evidence quality is rated as “very low” for most outcome measures, yet the conclusions are presented with high confidence.

3. The discussion highlights emodin’s potential but lacks depth regarding conflicting results or null findings.

---

## Round 0.2 · accepted · Accept

Thanks for addressing all the reviewers comments. Your manuscript can be accepted in its current form.

Reviewer 1 ·

Basic reporting

The manuscript is well-structured and adheres to the PeerJ standards for a systematic review and meta-analysis. The language used is clear and professional, making the content accessible to an international audience. The introduction provides a comprehensive background on the role of emodin in treating type 2 diabetes mellitus (T2DM) and justifies the need for this study effectively. The literature is well-referenced, supporting the context and relevance of the research question. Figures and tables are of high quality, well-labeled, and adequately described, aiding in the understanding of the study results. Additionally, the raw data have been provided in accordance with PeerJ’s policy, which enhances the transparency and reproducibility of the study.

Experimental design

The study design is robust and aligns with the scope of the journal. The research question is clearly defined, relevant, and addresses an important gap in the current understanding of emodin's therapeutic potential in T2DM. The authors conducted a thorough search of multiple databases, ensuring a comprehensive collection of relevant studies. The use of SYRCLE’s risk of bias tool for assessing the quality of animal studies is appropriate and adds credibility to the findings. The methods are described in sufficient detail, allowing for replication by other researchers. The choice of outcome measures, including glucose and lipid metabolism indicators, body weight, and serum insulin levels, is relevant and effectively addresses the study objectives.

Validity of the findings

The findings of this meta-analysis are compelling and demonstrate significant potential for emodin in treating glycolipid metabolism disorders. The data are robust, statistically sound, and appropriately controlled. The authors have effectively used RevMan and STATA software for data analysis, and the results indicate that emodin significantly reduces key indicators of glucose and lipid metabolism disorders in animal models of T2DM. The conclusions drawn are well-supported by the data and are directly linked to the original research question. The authors have also acknowledged the limitations of the study, including significant heterogeneity among the included studies, and have appropriately discussed the potential impact of these limitations on the results.

Additional comments

Overall, this manuscript provides a valuable contribution to the field of diabetes research. The systematic review and meta-analysis approach used by the authors is rigorous and provides a comprehensive overview of the current evidence on emodin's therapeutic effects. The findings suggest that emodin could be a promising candidate for further clinical investigation in the management of T2DM. The manuscript is well-written, and the study design and analysis are of high quality. I recommend this manuscript for publication in its current form.